# Hazardous Heavy Metals Accumulation and Health Risk Assessment of Different Vegetable Species in Contaminated Soils from a Typical Mining City, Central China

**DOI:** 10.3390/ijerph18052617

**Published:** 2021-03-05

**Authors:** Zhen Wang, Jianguo Bao, Tong Wang, Haseeb Tufail Moryani, Wei Kang, Jin Zheng, Changlin Zhan, Wensheng Xiao

**Affiliations:** 1School of Environmental Studies, China University of Geosciences, Wuhan 430074, China; wz5781@126.com (Z.W.); wangtong_1013@126.com (T.W.); haseebmoriani@cug.edu.cn (H.T.M.); 2Hubei Key Laboratory of Mine Environment Pollution Control and Remediation, School of Environmental Science and Engineering, Hubei Polytechnic University, Huangshi 435003, China; zhengjin0202@126.com (J.Z.); chl_zhan@126.com (C.Z.); wsxiao2002@126.com (W.X.)

**Keywords:** heavy metals, vegetables and crops, mining and smelting areas, health risk

## Abstract

Heavy metal poisoning has caused serious and widespread human tragedies via the food chain. To alleviate heavy metal pollution, particular attention should be paid to low accumulating vegetables and crops. In this study, the concentrations of five hazardous heavy metals (HMs), including copper (Cu), chromium (Cr), lead (Pb), cadmium (Cd), and arsenic (As) were determined from soils, vegetables, and crops near four typical mining and smelting zones. Nemerow’s synthetical pollution index (P_n_), Potential ecological risk index (RI), and Geo-accumulation index (I_geo_) were used to characterize the pollution degrees. The results showed that soils near mining and metal smelting zones were heavily polluted by Cu, Cd, As, and Pb. The total excessive rate followed a decreasing order of Cd (80.00%) > Cu (61.11%) > As (45.56%) > Pb (32.22%) > Cr (0.00%). Moreover, sources identification indicated that Cu, Pb, Cd, and As may originate from anthropogenic activities, while Cr may originate from parent materials. The exceeding rates of Cu, Cr, Pb, Cd, and As were 6.7%, 6.7%, 66.7%, 80.0%, and 26.7% among the vegetable and crop species, respectively. Particularly, vegetables like tomatoes, bell peppers, white radishes, and asparagus, revealed low accumulation characteristics. In addition, the hazard index (HI) for vegetables and crops of four zones was greater than 1, revealing a higher risk to the health of local children near the mine and smelter. However, the solanaceous fruit has a low-risk index (HI), indicating that it is a potentially safe vegetable type.

## 1. Introduction

Heavy metal (HM) contamination, including copper (Cu), chromium (Cr), lead (Pb), cadmium (Cd), and arsenic (As), is primarily attributed to anthropogenic activities, such as mining and metal smelting, traffic and factory emissions, ore exploitation dust fall, and sewage irrigation [1,2,3,4,5]. The soil quality of arable and cultivated lands near industrial and abandoned multiple metal mine is concerning. As reported by the Ministry of Environment Protection (MEP, China, 2014) [6], about 16.1% of total soils, including 19.4% of the arable lands and 36.3% of enterprise surrounding soils was over the Grade II environment quality standard (GB15618-1995) [7]. According to previous studies, agricultural soils were mainly polluted by anthropogenic activities such as mining exploitation, metal smelting, industrial processes, traffic emissions, and agricultural pesticides and fertilizers [8,9,10,11]. The soil-plant-human route is a dominant pathway for HMs in vegetables and crops. So far, many studies have shown that plant roots can uptake and accumulate these contaminants and gradually transfer them to the edible parts, ultimately posing a potential risk to human health through the food chain [12,13,14,15]. Besides, direct foliar uptake has been also considered as the direct pathway of atmospheric HMs such as Pb, Hg, and partly Cd which can be absorbed by stomata pores of leaves [16]. Therefore, the direct consumption of leaf vegetables can cause a severe threat to the health of inhabitants [17,18]. Some essential nutrients stored in the body are gradually depleted due to the consumption of foods containing HMs. Chronic exposure to Pb, Cd, and As can lead to various disorders, such as nervous system symptoms, kidney dysfunction and liver damage, lung and skin cancer, and bone fractures [3,19].

It is noteworthy that, currently, the Chinese government is paying considerable attention to food security, especially the concentration of HMs in vegetables and crops [12,20]. Many studies have investigated the heavy metal pollution of vegetables near tungsten mining in Gannan of Jiangxi [21]; Zhuzhou smelter of Hunan [22]; Jijie Pb/Zn smelter of Yunnan [23]; Tonglushan copper ming of Hubei in China [3], and other countries [24,25,26,27]. These findings revealed that most of the vegetables in the exposed areas exceed the maximum permissible concentration, and even caused a higher health risk than those in non-contaminated areas. It was reported that not only could the vegetables uptake amounts of HMs, but also elevated contents of HMs were found in food crops such as rice [12,28], wheat [29], and maize [30]. Particularly, Pb and Cd exhibited greater transfer capacity and ultimately accumulated in the grain [16,18]. However, most of the studies only considered a limited number of species in vegetables or certain hazardous metals. Little attention has been paid to the health risk of dietary intake to multiple hazardous metals. In addition, researchers have gradually drawn attention to how to limit HMs content in edible parts [31,32,33]. To move this issue, a new collection of low accumulating HMs species has been proposed. This ensures a reasonably low concentration of HMs in edible parts of vegetables or crop plants, even though they are cultivated on contaminated soil. Thus, this is one available pathway to reduce health risk, which may be due to the accumulation variation of different species and genotypes [33].

Daye City has a mining history of over 3000 years and has played a significant role in the national economic development of China as one of the largest copper producing areas. According to the National Ministry of Agriculture’s Environmental Protection Research and Monitoring Institute, heavy metal contamination of arable soils was extremely serious and mostly spread in villages and towns including Luojiaqiao, Jinhu, Dajipu, Chengui, and Jinshandian [34]. Previous studies have shown that Daye City’s soils, waters, and agricultural products have been polluted to higher degrees due to a vast number of industrial activities [3,34,35]. However, heavy metal exposure in agricultural soils typically occurs from a number of causes, not just mining activities. To date, studies to identify the sources of HMs mainly focused on a specific contaminated region or a small range of mining and smelting areas [16,21]. There are few studies on possible sources of HMs in Daye City [35]. Therefore, it is important to investigate some representative sites near the sensitive zones.

The main objectives of this study are to: (1) estimate heavy metals pollution levels and relevant impacts of soils, vegetables, and crops; (2) identify the sources of HMs in different study zones; (3) compare the multiple hazardous metals accumulation in vegetables and crops for the selection of low accumulating species; and (4) assess the dietary intakes and health risks of HMs in different food types to local children. The evaluated results will provide a scientific basis and theoretical support for environmental management in a typical mining city. Furthermore, the results also ensure human health and food safety and will provide valuable technical guidance to agricultural planting pattern.

## 2. Materials and Methods

### 2.1. Study Areas

The study areas (E 114°49′13″–114°57′9″, N 30°4′13″–30°13′29″) are located in Daye City which is situated in the middle reaches of the Yangtze River and in the southeast of Hubei Province, central China. The climate is the subtropical monsoon, and the average annual temperature is 17 °C. In addition, the average annual precipitation is 1382.6 mm with abundant rainfall. The four distinct seasons and superior water-heat conditions are conducive to the growth of crops.

Considering the relationship between heavy metals distributions and anthropogenic emissions, the study areas can be classified into four categories of vegetable fields near the non-ferrous metals smelter (A zone), abandoned copper mine (B zone), limestone quarry (C zone), and iron mine (D zone) (Figure 1). Moreover, they are located in four townships: Xialu District, Jinhu town, Huandiqiao town, and Jinshandian town. A zone is located in Xialu District, whose north part is adjacent to the open pit and concentrator. B zone is located in Jinhu town, which has a mining area of 7.8 km^2^ and a long history of open-cast mining. C zone is located at the eastern foot of Taiping Mountain of Huandiqiao town, which has been affected by quarrying and transportation for years. D zone is located in Jinshandian town, which is one of Wuhan Iron and Steel (Group) Company’s large underground iron mines.

### 2.2. Vegetable Samples and Rhizosphere Soils Collection

Vegetable and crop samples and rhizosphere soil samples were collected from April to October 2019. Ninety sampling sites were randomly selected near the mining areas. Each sampling site was recorded by using a global position system (GPS) (Figure 1). Ninety samples represented fifteen common species of vegetables, including four species of solanaceous fruits, five species of leafy vegetables, two species of rhizome geophytes, Legumes, and one species of nuts and cereals (Table 1), respectively. At each site, five sub-samples were collected and mixed to form an about 1.0 kg of the cross-sectional sample [27].

For each vegetable or crop sample at its corresponding site, we collected rhizosphere soil sample. In order to obtain a representative sample, five sub-samples were randomly collected at each site and bulked together to form a composite sample [3]. Approximately 500 g of soil samples were stored and transported to the laboratory immediately.

### 2.3. Sample Preparation and Analysis

The soil samples were air-dried for two weeks at room temperature, then grounded and sieved through a 100-mesh polyethylene sieve. All ground soil samples were stored in air-tight plastic bags before analysis. Each soil sample (~0.25 g) was prepared in a digestion tube with 10 mL of optimal levels of acid (HCl/HNO_3_/HF = 6/2/2). Then, they were heated at 190 °C for 30 min by using a microwave digestion oven (Mars5, CEM Corporation, Matthews, NC, USA) [36]. After cooling, the digested solution was filtered and diluted to 50 mL with 0.2% HNO_3_.

The vegetable samples were thoroughly washed by using tap water and ultrapure water and then weighed. After blanching at 105 °C for 30 min, samples were oven-dried at 60 °C to a constant weight for obtaining water percent, then ground to <1.0 mm by using a grinder. Each ash sample (0.5 g) was obtained using muffle furnace at 600 °C for 5 h, digested with 10 mL of ultrapure HNO_3,_ cooled and diluted with deionized water to 50 mL [3].

Soil pH levels were determined in a water/soil suspension with a ratio of 2.5:1 (*v*/*w*) [36]. The concentrations of heavy metal in soil and vegetable samples were determined by using inductively coupled plasma optical emission spectrometry (ICP-OES 7300, Thermo Fisher Scientific, Waltham, MA, USA). In order to ensure analytical precision and accuracy, sample replicates, reagent blanks, and standard reference materials (GBW 08301 and GBW 10015) from the Chinese Academy of Measurement Sciences were carried out simultaneously in each batch. The element recovery rate of 89–108% showed better accuracy. The relative standard deviations (RSD) were less than 5%. The limit of detection (LOD) for Cu, Cr, Pb, Cd, and As are 0.05, 0.05, 0.02, 0.002, and 0.002 mg/kg, respectively.

### 2.4. Data Analysis

#### 2.4.1. Nemerow’s Synthetical Pollution Index (Pn)

The level of soils contamination was assessed by the single-factor (Pi) and Nemerow’s synthetical pollution index (Pn) [37]. These indices were computed by Equations (1) and (2):(1)$$P_{i}=\sqrt{C_{i}/S_{i}}$$
where P_i_ is the single-factor index; C_i_ is the measured concentration of each heavy metal (mg/kg), and S_i_ is the standard value of each element (mg/kg). In this study, the standard of GB15618-2018 was applied as agricultural soil assessment criteria.
(2)$$P_{n}=\sqrt{\left(P_{\mathrm{imax}}^{2}+{\bar{P}}^{2}\right)/2}$$
where Pn is Nemerow’s synthetical pollution index, $\bar{P}$ is the average of single-factor index, $P_{\mathrm{imax}}$ is the maximum of single-factor index, and P_n_ is classified as safety (P_n_ ≤ 0.7, Class I), precaution (0.7 < P_n_ ≤ 1.0, Class II), slightly polluted (1.0 < P_n_ ≤ 2.0, Class III), moderately polluted (2.0 < P_n_ ≤ 3.0, Class IV), and seriously polluted (P_n_ > 3.0, Class V).

#### 2.4.2. Potential Ecological Risk Index (RI)

The potential ecological risk index (RI) was conducted to evaluate multiple HMs risk from soils, the index is a comprehensive consideration of HMs contents, environmental influence, and biotoxicity [38]. The RI is computed by Equations (3) and (4):(3)$$E_{r}^{i}=T_{r}^{i}\times C_{r}^{i}=T_{r}^{i}\times C_{s}^{i}/C_{n}^{i}$$
(4)$$\mathrm{RI}=\sum_{i=1}^{n}E_{r}^{i}$$
where $E_{r}^{i}$ represents the potential ecological risk index of a certain element, $C_{s}^{i}$ is the actual concentration of each heavy metal, $T_{r}^{i}$ is the toxic response coefficients and the values of Cu, Cr, Pb, Cd, and As are 5, 2, 5, 30 and 10, respectively. The RI classifications are low risk (RI < 150), moderate risk (150 ≤ RI < 300), considerable risk (300 ≤ RI < 600), and high risk (RI ≥ 600). The single element risk degrees ($E_{r}^{i}$) are low ecological risk ($E_{r}^{i}$ < 40), moderate ecological risk (40 ≤ $E_{r}^{i}$ < 80), considerable ecological risk (80 ≤ $E_{r}^{i}$ < 160), high ecological risk (160 ≤ $E_{r}^{i}$ < 320), and very high ecological risk ($E_{r}^{i}$ ≥ 320).

#### 2.4.3. Geo-Accumulation Index (Igeo)

The geo-accumulation index (I_geo_) was applied to evaluate spatial distribution pollution levels of different heavy metals in the four research areas, the index was calculated using the following Equation (5) [39]:(5)$$I_{\mathrm{geo}}=\log_{2}\left(C_{n}/K\cdot B_{n}\right)$$
where C_n_ is the concentrations of heavy metals (mg/kg), B_n_ is the soil background values in Hubei province (Table 1), K is introduced as the constant 1.5 to minimize some possible influence of background values due to geogenic variation. The contamination degrees of I_geo_ are classified as unpolluted (I_geo_ ≤ 0), slightly to moderately polluted (0 < I_geo_ ≤ 1), moderately polluted (1 < I_geo_ ≤ 2), moderately to heavily polluted (2 < I_geo_ ≤ 3), heavily polluted (3 < I_geo_ ≤ 4), heavily to extremely polluted (4 < I_geo_ ≤ 5), and extremely polluted (I_geo_ > 5).

#### 2.4.4. Health Risk Assessment

The health risk of heavy metals via vegetable consumption can be evaluated by target hazard quotients (THQ) and hazard index (HI) which was proposed by the US Environmental Protection Agency (USEPA 2000). The THQ was calculated using Equation (6) to assess the non-carcinogenic risks of individual heavy metal for local children. Moreover, The HI was computed to estimate the potential risk of multiple heavy metals by using the following Equation (7):(6)$$\mathrm{THQ}=\frac{\mathrm{EF}\times \mathrm{ED}\times C_{i}\times F_{\mathrm{IR}}}{\mathrm{RfD}\times \mathrm{BW}\times \mathrm{AT}}$$
(7)$$\mathrm{HI}=\sum_{n=1}^{i}{\mathrm{THQ}}_{n};i=1,2,3\ldots n$$
where C_i_ is the concentration of heavy metals in the edible part of vegetables (mg/kg, Fresh weight); F_IR_ is the ratio of food-intake for which the value of local children was 94 g/d [40]; EF is exposure frequency (365 d/a); ED is exposure duration (76.4 years, life expectancy in China according to World Health Statistics 2018); BW represents the average of body weight (18.7 kg for an adolescent); AT represents the average exposure time for non-carcinogenic effects (assuming 76.4 years, 365 days a year); RfD is the daily reference dose of heavy metals (μg/(kg·d)), the values of RfD for Cu, Cr, Cd, As were 40, 3.0, 1.0, 0.3, respectively [41]. Although Pb element had not been required in USEPA, Pb could exert a bad influence on the central nervous system of adolescents. Thus, the RfD for Pb was 3.5 μg/(kg·d) in present study [22].

If the value of THQ or HI is ≤1.0, there is no obvious health risk to residents; on the contrary, the non-carcinogenic risks are likely to occur when THQ or HI is >1.0, and the human health risk increases with the increase of THQ or HI.

#### 2.4.5. Statistical Analysis

Excel 2007 (Microsoft Office, Las Vegas, NV, USA), SPSS 23.0 (IBM Corporation, New York, NY, USA), and Origin 8.5, 2020b (OriginLab Corporation, Northampton, MA, USA) were used to conduct experimental data processing and statistical analysis. The Kolmogorov–Smirnov (K-S) test was used to determine whether the HMs’ contents were normally distributed in all areas [42]. A significant variance analysis was employed by using a one-way ANOVA test (LSD at *p* < 0.05 and *p* < 0.01). Pearson correlation analysis was used to clarify the relationships between total heavy metal concentrations and vegetable species in the different sites. Principal component analysis (PCA) and cluster analysis (CA) were conducted to identify possible sources in the study areas. ArcGIS 10.6 software (Environmental Systems Research Institute, Redlands, CA, USA) was used to evaluate spatial distribution levels with I_geo_ values.

## 3. Results

### 3.1. Heavy Metal Concentrations in Soils

The concentrations of HMs in soils are shown in Table 2. The pH of the soil in the four study areas showed a distinct pattern, with the mean values increasing in order as follow: D zone (pH = 5.05) < A zone (pH = 6.71) < B zone (pH = 7.03) < C zone (pH = 7.72). Most of the results of K-S test confirmed that the concentrations of HMs in the four zones passed the K-S test for normality (*p* > 0.05). Except Cr, all the soil HMs contents were higher than the background values of soil in Hubei Province. Compared with the soil environmental quality standard of agricultural land (GB 15618-2018), agricultural soils in the exposed zones were identified as contaminated areas by Cu, Cd, As and Pb, so the total excessive rate of them followed a decreasing order of Cd (80.00%) > Cu (61.11%) > As (45.56%) > Pb (32.22%) > Cr (0.00%).

In the A zone, the mean contents of Cu and Cd were significantly higher than the national standard values in all sampling sites, and the excessive rate followed the order of Cu (100%) = Cd (100%) > As (71.4%) > Pb (61.9%) > Cr (0.0%). In the B zone, the excessive rate values of HMs decreased in the following order of Cu (100%) > Cd (88.46%) > As (53.85%) > Pb (42.31%) > Cr (0.0%). In the C zone, the excessive rate of HMs increased in the following order of Cu (30.0%) < As (60.0%) < Cd (85.0%). However, in the D zone, only the mean contents of Cu, Pb, and Cd exceeded the soil standard with the excessive rates of 8.7%, 21.74%, and 47.83%, respectively. Compared to the four zones, almost all the HMs concentrations of soil near the non-ferrous metals smelter, abandoned copper mine and limestone quarry were significantly higher (*p* < 0.05) than those of iron mine, especially Cu, Cd, and As showed more extreme levels of pollution. Furthermore, the coefficients of variation (CV%) for Cu, Cd, Pb, and As were over 50% in the four study areas, indicating that a larger influence of anthropogenic activities contributed to the heavy metals in the soils.

### 3.2. Contaminated Evaluation of HMs in Soils

#### 3.2.1. Nemerow’s Synthetical Pollution Index (Pn) and Potential Ecological Risk (RI)

In order to obtain an accurate contaminated status of different zones, the values of the single-factor index and Nemerow’s synthetical pollution index were calculated (Appendix A). The most soil samples were polluted to varying levels ($p_{i}\;$> 1) except for Cr. The results indicate that the A and B zones were highly polluted by Cd ($P_{i}\;$= 12.98) and Cu ($P_{i}\;$= 12.25). As illustrated in Figure 2a, the $P_{n}$ values ranged from 0.56 to 18.44, exposing multiple ranges from safety to serious pollution. Thus, the pollution index ($P_{n}$) of four regions decreased in the following order: A zone > B zone > C zone > D zone.

As shown in Figure 2b, the higher potential ecological risk was found in A and B zones in comparison with C and D zones. More specifically, in the A zone, the single ecological risk index ($E_{r}^{i}$) of Cd was higher than 320 ($E_{r}^{i}$ = 389.4), indicating a higher contribution of Cd to potential ecological risk (Appendix A). In the B zone, except Cd, the $E_{r}^{i}$ value of Cu was larger than 40 ($E_{r}^{i}\;$= 63.0), posing a moderate potential ecological risk though RI being lower risk level than non-ferrous metals smelter (Appendix A). In the C zone, only the single ecological risk index of Cd was more than 40 ($E_{r}^{i}\;$= 55.2), suggesting a moderate potential ecological risk (Appendix A), while D zone corresponded little sensitivity of biological community to HMs in soils because all the $E_{r}^{i}$ values were less than 40 except Cd (Appendix A). Based on the results of $E_{r}^{i}$ and RI in the study areas, Cd and Cu in four categories of vegetable garden soil exhibited higher ecological risk.

#### 3.2.2. Geo-Accumulation Index (Igeo)

The spatial distributions of I_geo_ values of HMs in soils are shown in Appendix A. In this study, we found that the I_geo_ values were greater than zero to some degree in the three different categories of vegetable fields excluding the D zone. Among the five hazardous metals, the pollution levels of I_geo_-Cd showed larger spatial variation in Figure 3, suggesting that different anthropogenic activities can contribute to its distributions. In contrast, the negative I_geo_ value of Cr is observed in Appendix A, the concentration of Cr was obviously below the background values, which indicated the study area free from Cr contamination.

### 3.3. Heavy Metals in Vegetables and Crops

The concentrations of HMs in edible parts of vegetables and crops in the study areas are presented in Appendix A. The mean contents of Cu, Cr, Pb, Cd, and As from vegetables and crops were 4.05, 0.22, 0.41, 0.15, and 0.26 mg/kg (FW) in the exposed areas (A–C zones), respectively, which was 2–8 folds higher than those in D zones (*p* < 0.05). Additionally, several larger coefficients of variation (CV% ≥ 100%) of HMs were displayed in the study areas, suggesting that a wide variation of values could be related to the difference of vegetable species.

In present study, the maximum allowable concentration (MAC) of food standard (GB2762-2017) is presented in Appendix A. Compared to 15 species of vegetables and crops, the highest Cu contents were found in brassica campestris (Figure 4a), which almost surpassed the MAC value, while the higher Cr content was determined in peanut (Figure 4b). Moreover, the species exceeding rates of Cu and Cr were 6.7%. According to the MAC, Pb and Cd contents in most leaf vegetables were found above the criterion line as shown in Figure 4c,d, whereas relatively low concentration was found in tomato, bell pepper, white radish, and asparagus bean. In contrast, several crop types including peanut, ipomoea, maize and soya bean showed the higher accumulation ability for Pb and Cd. So, the percentage of vegetables and crops that exceed the MAC value is 66.7% for Pb and 80.0% for Cd. For As, only 26.7% species exceed the MAC, and the contents of As in the vegetables are listed in Figure 4e, the higher value can be observed in leaf vegetables, especially in edible amaranth.

Overall, in tomato, bell pepper, white radish, and asparagus bean, five hazardous metals contents were rarely observed, indicating low accumulation characteristics to multiple hazardous metals. Furthermore, the observed HMs concentrations of all collected samples in the contaminated areas (A–C zones) were significantly higher than those in D zone.

### 3.4. Source Analysis of HMs in Different Contaminated Soils

Pearson correlation analysis was applied to clarify the relationship of soil HMs in the contaminated zones. As shown in Figure 5, the total amount of several HMs in soils showed a significantly positive correlation between pairs of elements in the four study areas. The correlation coefficients of Cu-Pb, Cu-Cd, Cu-As, Pb-Cd, Pb-As, and Cd-As in the A zone were 0.91, 0.76, 0.87, 0.75, 0.89, and 0.77 (*p* < 0.01), respectively. Furthermore, the correlation coefficients of Cu-Cr and Pb-Cd in the B zone were 0.64 and 0.68 (*p* < 0.01), Cr-Pb and Pb-As were 0.46 and 0.47 (*p* < 0.05), respectively. The positive correlation coefficients of Cu-Pb (0.72) and Cd-As (0.45) were also observed in the C zone, and Cu-Cr (0.91) was significantly correlated in the D zone, which indicated that these elements were derived from the same sources. Nevertheless, the negative correlation coefficients of Cr-Cd, Cr-As, and Pb-Cd in C and D zones were calculated to be −0.56, −0.49, and −0.48, which suggested that these elements could originate from different sources. As shown in Appendix A, the correlation coefficients between soils and foodstuffs were not found at significant range (*p* < 0.05), indicating that they were barely or negatively correlated with each other. The possible explanation is that the enrichment of HMs in foods is affected not only by soil HMs, but also by vegetable and crop species or other complex factors [22,29].

Principal component analysis (PCA) and cluster analysis (CA) were used to further identify the sources among the five heavy metals in the study areas. The results of the PCA are shown in Appendix A. Two principal components were extracted in the four study zones because the eigenvalues were greater than 1. In the A zone, PC1 explained 69.70% of the variance and positively related to Cu, Pb, Cd, and As, while PC2 accounted for 22.13% of the variance, which was dominated by Cr. In the B zone, PC1 explained 49.13% of the variance and depended partially on Pb, Cd, and As; meanwhile, PC2, which was dominated by Cu and Cr, accounted for 23.14% of the variance. In the C zone, PC1 occupied 45.50% of the variance and loaded positively on Cd and As, additionally, PC2 accounted for 29.73% of the variance and was dominated by Cu and Pb. In the D zone, PC1 occupied 43.67% of the variance and depended positively on Cu and Cr, while PC2 accounted for 27.52% of the variance and was dominated by Pb. Based on the results of the previous spatial distribution, the concentrations of several HMs (Cu, Pb, Cd, and As) were higher than the soil background values of Hubei province. Thus, the PCA results revealed that these HMs were derived from anthropogenic activities, Cr could originate from natural process.

The dendrogram results obtained from CA for HMs are shown in Appendix A. Two main clusters were distinguished as following: (1) Cu-Pb-Cd-As, (2) Cr in the A and C zone; (1) Pb-Cd-As, (2) Cu-Cr in the B and D zones, respectively. It revealed similar results as Pearson correlation and principal component analysis.

### 3.5. Health Risk Assessment of Dietary Intake

The HI and THQ were applied to assess the non-carcinogenic risk of food intake which was considered as the main exposed risk pathway. In Appendix A, the HI values for all zones were over 1, indicating a higher non-carcinogenic risk of dietary intake for local children. These findings are consistent with the results of earlier research [31,40]. In addition, the HI values of four investigated zones followed a decreasing order: B zone > A zone > C zone > D zone. As shown in Figure 6, the THQ values of solanaceous fruits were lower than those of the other five types. In contrast, the highest HI values were observed in leaf vegetables posing the highest potential health risk for local children.

The health risk contribution rates of different elements in the four zones are shown in Appendix A. Among the five hazardous metals, As was the primary contributor for health risk in all the four zones, and the result of contribution rate were 67.71% to A zone, 72.31% to B zone, 67.87% to C zone, and 43.48% to D zone.

## 4. Discussion

### 4.1. Pollution and Source Identification in Soils

As numerous studies reported, heavy metal pollution of soil is a severe environmental concern, especially in areas where agricultural soils are close to anthropogenic pollution, such as mining and smelting zones [1,8], industrial suburban areas [43], e-waste dismantling areas and traffic roadsides [16,44,45]. Our results indicated that the contents of Cu and Cd in all sites near the Tonglushan mine and copper smelter were lower than those in previous studies [40]. The level of soil pollution is affected by source distance, Li et al. [46] and Sun et al. [47] also presented that sampling sites closer to the mine were more seriously polluted by Cu, Cd, Pb, and As, and as expected, the vegetables and crops were also contaminated by these toxic metals to a greater extent, which threatened the health of residents. In addition to the variation of sampling sites, the possible reason is that these toxic metals, especially Cu and Cd, are easily absorbed by crop uptake or phytoextraction from soils. Furthermore, the contents of Cd and Cu in current study were significantly higher than those of other areas in Daye City (Cd: 1.41 mg/kg and Cu: 105 mg/kg) [35]. The pollution of HMs was serious in the agricultural soils, so the study zones chosen could be more vulnerable areas impacted by mining and smelting. In comparison with other countries, the concentrations of Cu, Pb, and Cd were much higher than those in Bangladesh [4]. However, in Germany and Northern Tunisia, the soil Pb concentration was 10-fold higher than that in this study [24,26].

The four study zones are representative areas polluted by HMs from different anthropogenic activities. In the A zone, an elevated concentration of Cd was observed in soil-vegetable system. Many studies showed that non-ferrous smelter could lead to a large amount of Cd emitting into the soils through rainfall and deposition [48]. According to this study, Cu, Pb, Cd, and As may originate from smelting activities. Bi et al. [16] also reported that the Cd, Pb, Cu, and As contents in soil and road dust exceeded the referenced values near industrial facilities in Shanghai, suggesting that one of the major sources of HMs pollution were industrial sources, including metal and metal compound application. It is evident in the B zone that mining operations have polluted vast fields of agricultural land. Pb, Cd, and As are known to be the associated elements of Cu, and primarily derived through waste water discharge, spoil disposal, and dust fall from copper mining processes. In the C zone, enrichment of As can be related with agricultural activities, as the superfluous use of pesticides, herbicide, phosphate fertilizers, and livestock manure are also important sources for As in soil [49,50]. In addition, vehicle exhaust emissions and road dust can be considered as other major sources for Pb and Cd in soils [51,52]. Several cement plants and lime quarries were installed along the roadside so the contribution of transportation to Pb cannot be ignored. Similar findings were found in the D zone, the major sources of Pb could be considered as atmospheric deposition and automobile exhaust. According to PCA results, Cr may originate from natural sources related to parent materials. HMs, one of the massive rock elements, can be released into the environment by rock weathering, rain erosion and volcanic eruption. Cr, an important source of mineral element, can reach into soil-foodstuffs-human chain by radicular system absorption.

Based on the above mentioned argument, the distribution patterns of HMs in mining and smelting areas were similar to those in industrial areas, which were also regarded as the most sensitive regions for HMs contamination. Due to multiple sources of heavy metals emission in different regions, it is difficult to identify the source of one heavy metal pollution. Yet we might at least find a great link between emissions of heavy metals and anthropogenic sources.

### 4.2. Comparisons of HMs Accumulation in Different Vegetable and Crop Species

Previous studies have reported that elevated concentration of HMs was found near the multiple-metal mining and smelter, e-waste dismantling area, and industrial sites [16,23,45,53]. In addition, Rahmdel et al. [17] and Mandal et al. [20] reported that higher concentrations of Cu and Pb in leaf vegetables were observed in Iran and India than those in the present study, in which samples were collected near the rural districts of industrial town. Comparatively, in China, the average concentrations of HMs in vegetables were mostly similar to those in the peri-urban areas, but most of them were significantly higher than those from Pb/Zn smelters in Gejiu of Yunnan province [23], and seven areas in Zhuzhou of Hunan Province [22]. Therefore, we inferred that this was possibly linked to the industrial activities of local copper mining and smelter operations in Daye City. For over 3000 years, the Tonglushan Mine has become an inactive copper mine. Moreover, the Daye Nonferrous Metal Smelting Plant has also been releasing numerous toxic metals as by-products into the atmosphere and soil for almost 50 years. Along these lines, Cai et al. [40] measured the amount of contamination of food crops grown near the Tonglushan Mine of Daye City, which found that the contents of Cd and As in rice and most mine-affected vegetables were above the maximum permissible limits. Similar results also were obtained in Daye copper smelter. Du et al. [35] also indicated that in the vicinity of the active Pb-Zn mine and Cu smelter, a higher environmental risk can occur to rice and vegetable due to Cd exposure.

Referring to the sources of heavy metals in vegetables and crops, previous studies have revealed relationship between physiochemical properties of soil and the contents of HMs in foods. They have established certain migration and transport models for predicting bioaccumulation factors of HMs in soil-vegetable or soil-crop system [14,54]. In this study, five heavy metals, especially Pb, Cd, and As, were highly enriched in leaf vegetables, while barely accumulated in fruit vegetables collected from the heavy contaminated sites. Similar to our findings, Zhou et al. [55] and Hu et al. [56] also reported that leaf vegetables can uptake more heavy metals than rootstalk and fruit vegetables. Generally, the contents of Cd in leaf vegetables were nearly 3 times than those in non-leaf vegetables. By Comparing the contamination levels of vegetables in the open field with those in greenhouse, Li et al. [57] found that the accumulation of Cd in most vegetables was mainly from soil rather than atmospheric deposition. This result further confirmed that Cd is typically a labile trace element in the soil, and many vegetables, especially leaf vegetables, are more capable of absorbing it. The higher translocation rates of Cd were also found even when vegetables were grown in low-Cd soils [13,58]. Moreover, the accumulation of Cd in vegetables is closely related to the amount of soil extractable Cd depending primarily on the properties of the soil, such as pH, organic matter, cation exchange capacity and clay content [13,59,60]. However, this is difficult for Pb to migrate through the soil-to-crops due to its low solubility and bioavailability [30,61]. In order to explore the influence of airborne heavy metals on vegetable species, Harrison et al. [62] confirmed that Pb in the atmosphere contributed as much as 85% to spinach pollution, while Cd only contributed 23%. In addition, Bi et al. [16] further demonstrated that atmospheric Pb via foliar uptake is the primary source for the accumulation of Pb in leaf vegetables by using ^208^Pb/^206^Pb and ^207^Pb/^206^Pb ratios of isotopic composition. In terms of As accumulation, the higher concentration of As was found in leaf vegetables than fruits vegetables [34]. In general, Cr and Cu were readily absorbed by roots and their distribution was ordered as following root > stem > leaf [63]. Similarly, the higher concentrations of Cr were observed in rice and wheat [12]. However, our results indicated that the elevated Cr contents in ipomoea and peanut were accumulated mainly from soils, implying high potential health risk via the consumption of rhizome geophytes.

Vegetables and crops grown in contaminated farmlands could have higher accumulation capabilities for HMs than those cultivated in the safe areas [47]. However, we found that tomato, bell pepper, white radish, and asparagus bean have lower heavy metal contents than leaf vegetables, suggesting that these species showed low accumulation characteristics for hazardous heavy metals. In general, the accumulation of HMs in vegetables could be affected by types and species as well as spatial distribution of HMs in soils [64]. Low accumulation abilities may be governed by the expression of specific genes [33]. Previous studies have provided genetic basis for screening and breeding low accumulating species such as rice, wheat, Chinese cabbage, and tomato [31,33,65]. On the basis of the results of this research, it is believed that foodstuffs with low accumulation are found in contaminated soils, suggesting that these genotypes can be treated as reliable native cultivars.

### 4.3. Health Risk Assessment for Local Children

Contaminated soil particles and foods are the most important exposures for humans, which can lead to potential health risks for residents. Based on non-carcinogenic risk assessment proposed by USEPA [66,67,68], the average daily dose (ADD) was dependent on 3 factors including ingestion, inhalation, and dermal contact. Generally, children are more sensitive to hazardous elements via non-dietary ingestion than adults because of physiological activities, such as rubbing eyes and sucking fingers [37,69,70]. Furthermore, food ingestion was the leading contributor to children’s ADD of HMs in comparison with soil ingestion and dermal [71]. Similar findings have been demonstrated by Huang et al. [2] and Cai et al. [40]. In this study, local children living around mining and smelter areas could suffer from adverse health impacts because the HIs of leaf vegetables were higher than 1. Interestingly, the HI values of solanaceous fruits and legumes were less than 1, suggesting that the presence of low accumulation foods was healthy for local children.

Previous studies indicated the higher accumulation ability for Cu, Pb, and Cd in leaf, rhizome vegetables, and rice grain [2,22,64]. In this study, As had a higher THQ value than other hazardous heavy metals. A similar report also confirmed that As exposure through vegetable ingestion has significant adverse effects on residents [23]. Thus, much attention should be paid to As although there is a lack of detailed criteria of chemical form. It is a well-established fact that the availability and toxicity of HMs depend on speciation and chemical states of HMs. Such as, As (III), binding to certain enzymes and proteins from phytochelatins, is recognized as a more poisonous form than As (V), and it is readily absorbed via plant roots from the soil [72]. Dhanker et al. [72] documented that As (V) can be reduced into As (III) as a result of arsenate reductase inside most plant tissues. In general, detoxification of As can be achieved via methylation reaction with low toxic intermediate organic arsenic chemicals including monomethylarsonic acid (MMA), dimethylarsinic acid (DMA), and trimethylarsine oxide (TMAO) [73]. However, Zhou et al. [74] found that monomethylarsonic acid (MMAIII) and dimethylarsinic acid (DMAIII) in the human body may be more or less cytotoxic to As (III), which were recognized as human hepatocytes impairer.

Cr also exists in two common types of oxidation states including Cr (III) and Cr (VI). In comparison, Cr (VI) originates from some anthropogenic compounds in the processing of industry and smelter, which is the most toxic form due to higher solubility, mobility and oxidizability, thus Cr (VI) can enter into human body via food chain and eventually damage tissues. According to previous studies, the high Cr levels had been found in several foods, posing potential human health risk [75,76,77]. Due to the elevated toxicity in different chemical states, a strict permissible risk index for foods should be developed for metal states like Cr (VI) and As (III). To sum up, it is necessary to clear up the relationship between element chemical form and human health risk.

## 5. Conclusions

In this study, four typical vegetable fields around the non-ferrous metals smelter, abandoned copper mine, limestone quarry, and iron mine in Daye City were selected as study zones. The results showed that the pollution of non-ferrous metals smelter and abandoned copper mine was serious, and heavy metals pollution was mainly attributed to mining and metal smelting activities. The main polluted metals were followed a decreasing order of Cd (80.00%) > Cu (61.11%) > As (45.56%) > Pb (32.22%) > Cr (0.00%). According to the results of spatial distribution and PCA, these five elements could be classified into two groups: (1) PC1 (Cu, Pb, Cd and As) which may originate from anthropogenic activities; (2) PC2 (Cr) which was mostly associated with the parent material of soil. Compared with vegetable and crop species, the contents of HMs in tomato, bell pepper, white radish, and asparagus bean were relatively low or closed zero, possibly due to their low accumulation characteristics. However, the elevated contents of Cd, Pb, and As were observed in peanut, ipomoea, maize, soya bean, and most leaf vegetables.

In addition, the results of health risk assessment revealed that the HI values of four zones through consumption of vegetables and crops were >1, revealing that the local children near mining and smelter regions suffer a higher non-carcinogenic risk. In contrast, the consumption of solanaceous fruits could cause a low potential health risk, which suggested that low accumulation species should be selected and applied on contaminated agricultural soils. Thus, adjustment of planting structure or conversion of land use should be adopted to protect human health and ensure food safety.

## Figures and Tables

**Figure 1 ijerph-18-02617-f001:**
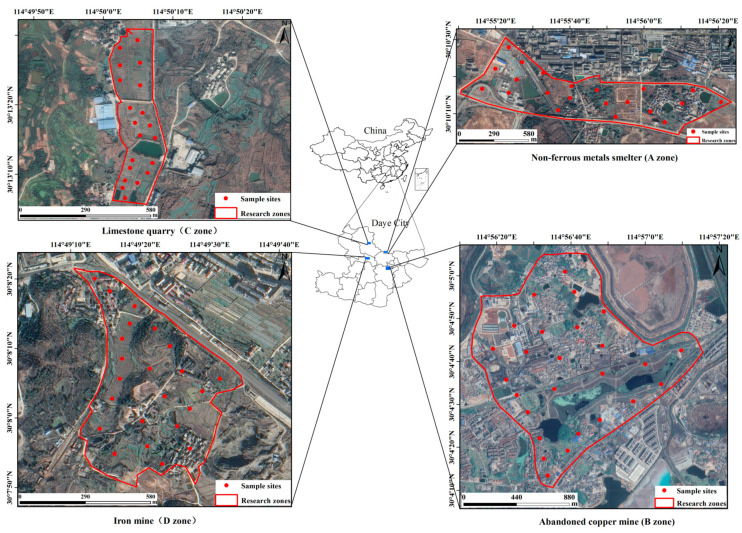
Locations of the investigated areas and sampling sites in Daye City.

**Figure 2 ijerph-18-02617-f002:**
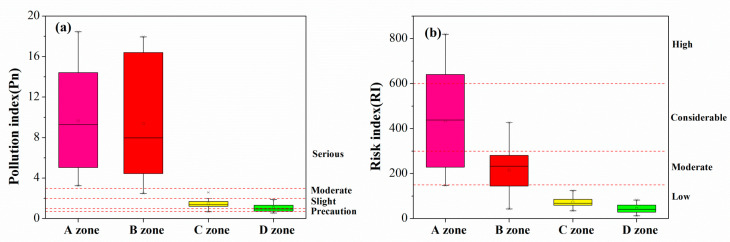
Boxplot of pollution index (**a**) and risk index (**b**) for heavy metals in the four zones.

**Figure 3 ijerph-18-02617-f003:**
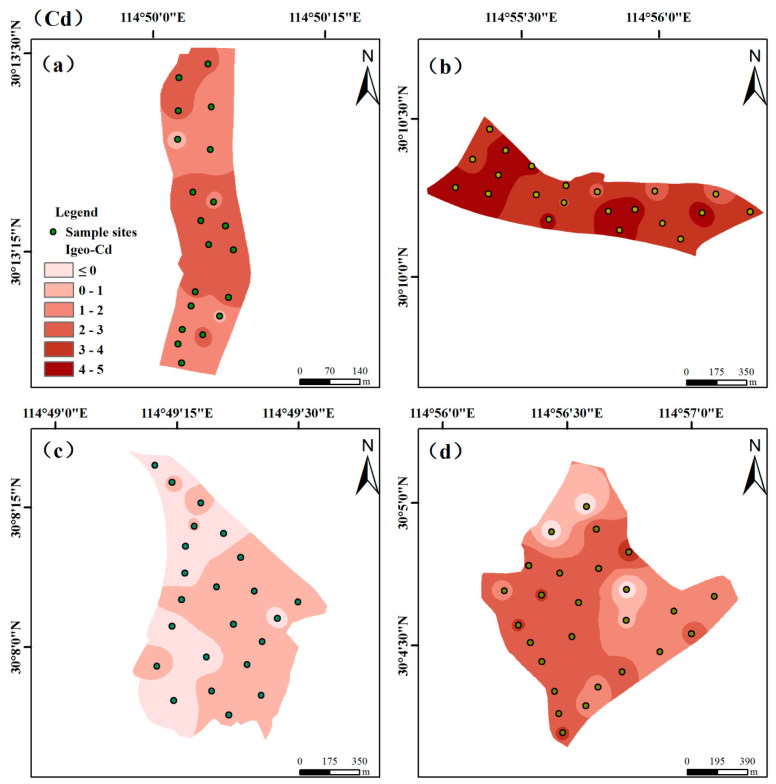
Spatial distribution pollution levels of Igeo-Cd in the four zones: (**a**) Non-ferrous metals smelter; (**b**) Abandoned copper mine; (**c**) Limestone quarry; (**d**) Iron mine.

**Figure 4 ijerph-18-02617-f004:**
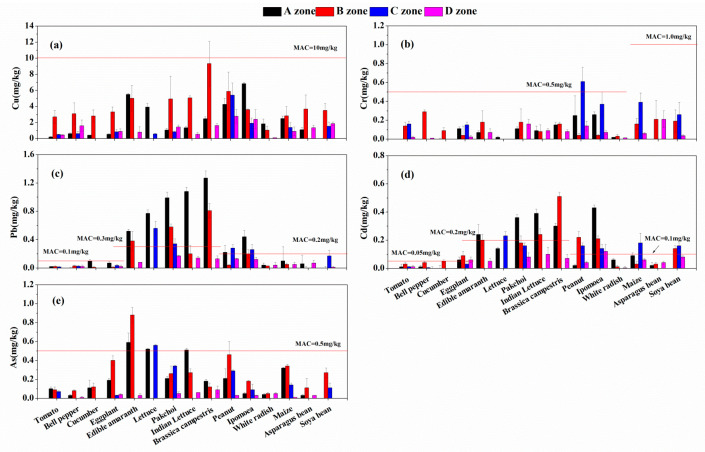
Heavy metals in edible parts of vegetable and crop species in the four zones: (**a**) Cu; (**b**) Cr; (**c**) Pb; (**d**) Cd; (**e**) As.

**Figure 5 ijerph-18-02617-f005:**
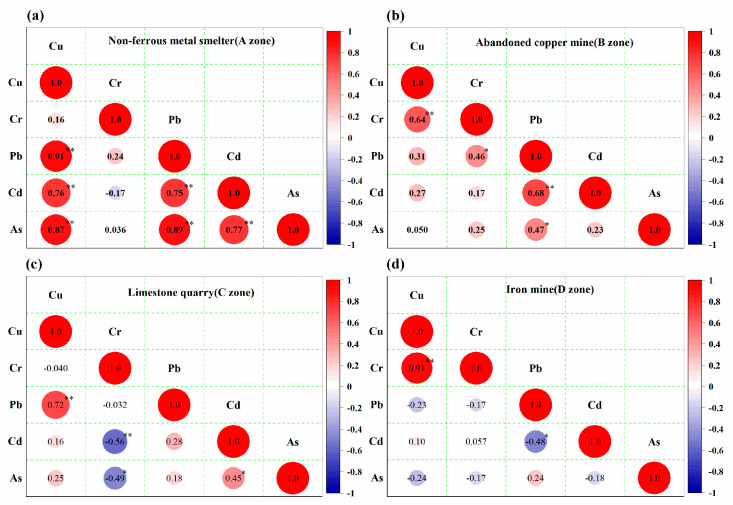
Correlation plots of heavy metals in the four zones: (**a**) Non-ferrous metals smelter; (**b**) Abandoned copper mine; (**c**) Limestone quarry; (**d**) Iron mine; * *p* < 0.05, ** *p* < 0.01.

**Figure 6 ijerph-18-02617-f006:**
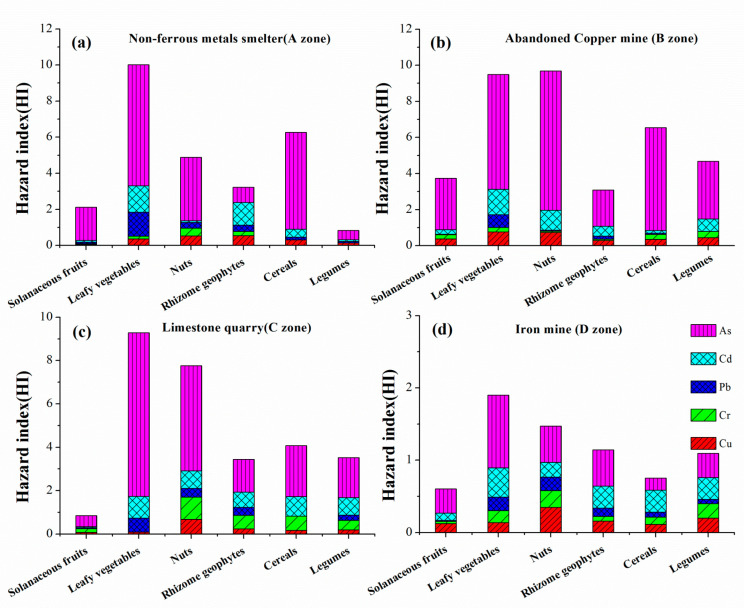
Target hazard quotients (THQ) and hazard index (HI) of different vegetable and crop types: (**a**) Non-ferrous metals smelter; (**b**) Abandoned copper mine; (**c**) Limestone quarry; (**d**) Iron mine.

**Table 1 ijerph-18-02617-t001:** Vegetable species used in the experiments.

NO.	Types	Vegetable Species	Harvest Season
1	Solanaceous fruits	Tomato (*Lycopersicum esculentum* Mill.)	Spring and summer
2		Bell pepper (*Capsicum annuum* L.)	Spring and summer
3		Cucumber (*Cucumis sativus* L.)	Spring and summer
4		Eggplant (*Solanum melongena* L.)	Spring and summer
5	Leafy vegetables	Edible amaranth (*Amaranthus tricolor* L.)	Spring and summer
6		Lettuce (*Lactuca sativa* L.)	Spring and summer
7		Pakchoi (*Brassica Chinensis* L.)	Spring and summer
8		Indian Lettuce (*Lactuca sativa* L. var longifoliaf. Lam)	Spring and summer
9		Brassica campestris (*Brassica campestris* L. var. purpurea Baileysh)	Spring and summer
10	Nuts	Peanut (*Arachis hypogaea* Linn.)	Autumn and winter
11	Rhizome geophytes	Ipomoea (*Ipomoea batatas* L.)	Autumn and winter
12		White radish (*Raphanus sativus* L.)	Autumn and winter
13	Cereals	Maize (*Zea mays* L.)	Autumn and winter
14	Legumes	Asparagus bean (*Vigna unguiculata* Linn.)	Spring and summer
15		Soya bean (*Glycine max* Linn.)	Autumn and winter

**Table 2 ijerph-18-02617-t002:** Characteristics of heavy metals in soils of the four typical zones.

Sampling Areas		pH	Cu (mg/kg)	Cr (mg/kg)	Pb (mg/kg)	Cd (mg/kg)	As (mg/kg)
Non-ferrous metals smelter (*n* = 21)	Range	6.51–6.87	116.17–858.97	61.33–94.82	27.04–219.51	0.44–7.51	4.49–75.54
	Mean ± SD	6.71 ± 0.13	475.39 ± 226.10	72.27 ± 7.55	136.89 ± 59.99	3.80 ± 2.10	42.48 ± 21.26
	CV/%	1.94	47.56	10.45	43.82	55.26	50.05
	K-Sp	—	0.200 *	0.200 *	0.006	0.190 *	0.144 *
	^a^ Excessive rate %	—	100.00	0.00	61.90	100.00	71.4
Abandoned copper mine (*n* = 26)	Range	6.58–7.35	213.36–1446.10	66.21–131.70	33.25–223.48	0.15–3.28	4.09–86.04
	Mean ± SD	7.03 ± 0.30	729.39 ± 383.83	95.69 ± 19.37	107.76 ± 56.64	1.34 ± 0.78	41.96 ± 21.68
	CV/%	4.25	52.62	20.24	52.56	58.21	51.67
	K-Sp	—	0.062 *	0.200 *	0.000	0.200 *	0.200 *
	^a^ Excessive rate %	—	100.00	0.00	42.31	88.46	53.85
Limestone quarry (*n* = 20)	Range	7.56–7.83	74.53–123.75	5.10–99.78	28.01–103.25	0.40–2.06	2.93–45.98
	Mean ± SD	7.72 ± 0.11	94.45 ± 14.93	55.00 ± 31.22	60.57 ± 19.77	1.10 ± 0.45	26.45 ± 11.20
	CV/%	1.41	15.81	56.76	32.64	40.91	42.34
	K-Sp	—	0.134 *	0.200 *	0.200 *	0.200 *	0.200 *
	^a^ Excessive rate %	—	30.00	0.00	0.00	85.00	60.00
Iron mine (*n* = 23)	Range	4.76–5.42	20.17–64.48	49.12–80.77	36.09–76.25	0.05–1.75	0.79–39.24
	Mean ± SD	5.05 ± 0.22	36.31 ± 11.02	58.23 ± 7.72	52.34 ± 13.53	0.45 ± 0.41	8.33 ± 9.49
	CV/%	4.29	30.35	13.26	25.85	91.11	113.93
	K-Sp	—	0.125 *	0.005	0.011	0.021	0.000
	^a^ Excessive rate %	—	8.70	0.00	21.74	47.83	0.00
^b^ Background values			30.7	86	26.7	0.17	12.3
^c^ Soil pollutant standards			50, 100	150, 200, 250	70, 90, 120, 170	0.3, 0.6	40, 30, 25
Total excessive rate %			61.11	0.00	32.22	80.00	45.56

K-Sp is the significance level of Kolmogorov-Smirnov test for normality, * *p* > 0.05. ^a^ Refer to soil pollutant standards. ^b^ Soil background values in Hubei province (Soil background values of Chinese elements 1990). ^c^ Soil environment quality: Risk control standard for soil contamination of agricultural land (GB15618-2018 Ministry of Environment Protection).

## Data Availability

No new data were created or analyzed in this study. Data sharing is not applicable to this article.

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
