# Peer review of "Hazardous Heavy Metals Accumulation and Health Risk Assessment of Different Vegetable Species in Contaminated Soils from a Typical Mining City, Central China"

_ijerph, 2021, doi:10.3390/ijerph18052617_

Round 1

Reviewer 1 Report

Dear authors

This is an interesting study. The paper is generally well written and structured. However, in my opinion the paper has some shortcomings. Below I have provided numerous remarks. Furthermore I made additional suggestions for more in-depth analyses of the data.

General suggestions:

  1. A table with vegetable species used in the experiments should be given in the text and not in the Appendix. The factor ‘species’ is a crucial one in these experiments.
  2. Sample preparation and analysis should be followed by references
  3. Table 1, Table A6 should be reformatted
  4. Why all the tables and figures that are mentioned in the text from the Appendix A are mentioned as S1…. And in Appendix A as A1. It is confusing…
  5. a lack of a discussion of effects of the distance

Other suggestions:

  1. line 35 (MEP, China, 2014) have to be included in references
  2. line 117 species and not specie
  3. line 292 why FW and not dry weight.
  4. line are there scientific data supporting “Cr could originate from natural process”. Maybe soil analyses?
  5. Line 447-449.In which way is this relevant with this study?
  6. Line 595 Figure A3 maybe?

Given these shortcomings the manuscript requires minor revisions.

Author Response

Dear reviewer,

Best

Reviewer 2 Report

The paper is related to the accumulation of heavy metals in soils and crops located in 4 regions near mining sites.

I find the document interesting and appropriate for the IJERPH journal, as it covers aspects related to environmental pollution and human health.

The document has a good writing and use of the English language.

The following is recommended to be considered for publication:

Reduce the length of the results sections. For example, in the subsection "Heavy metal concentrations in soils" it is a bit difficult to read, since the same comparison of heavy metal levels is made for all regions. It would be convenient to group these data in the form of a table and then mention the findings in the text.

The paper contains an appendix section where a large number of results are added, it is recommended to evaluate the relevance of adding all the results shown

The discussion section seems to me adequate, I would only suggest adding some studies with reference to contamination levels or international standards, not only those related to China.

Author Response

Dear reviewer,

Best

Reviewer 3 Report

Dear Authors,

Please find the comments in the attached file.

Best

Author Response

Dear reviewer,

Best
